# BACKDROP: STOCHASTIC BACKPROPAGATION

## ABSTRACT

We introduce backdrop, a flexible and simple-to-implement method, intuitively described as dropout acting only along the backpropagation pipeline. Backdrop is implemented via one or more masking layers which are inserted at specific points along the network. Each backdrop masking layer acts as the identity in the forward pass, but randomly masks parts of the backward gradient propagation. Intuitively, inserting a backdrop layer after any convolutional layer leads to stochastic gradients corresponding to features of that scale. Therefore, backdrop is well suited for problems in which the data have a multi-scale, hierarchical structure. Backdrop can also be applied to problems with non-decomposable loss functions where standard SGD methods are not well suited. We perform a number of experiments and demonstrate that backdrop leads to significant improvements in generalization.

## 1 INTRODUCTION

Stochastic gradient descent (SGD) and its minibatch variants are ubiquitous in virtually all learning tasks (Ian Goodfellow & Courville (2016); Bottou et al. (2018)). SGD enables deep learning to scale to large datasets, decreases training time and improves generalization. However, there are many problems where there exists a large amount of information but the data is packaged in a small number of information-rich samples. These problems cannot take full advantage of the benefits of SGD as the number of training samples is limited. Examples of these include learning from high resolution medical images, satellite imagery and GIS data, cosmological simulations, lattice simulations for quantum systems, and many others. In all of these situations, there are relatively few training samples, but each training sample carries a tremendous amount of information and can be intuitively considered as being comprised of many independent subsamples.

Since the benefits of SGD require the existence of a large number of samples, efficiently employing these techniques on the above problems requires careful analysis of the problem in order to re-structure the information-rich data samples into smaller independent pieces. This is not always a straight-forward procedure, and may even be impossible without destroying the structure of the data (Ravanbakhsh et al. (2017); Shanahan et al. (2018)).

This motivates us to introduce backdrop, a new technique for stochastic gradient optimization which does not require the user to reformulate the problem or restructure the training data. In this method, the loss function is unmodified and takes the entirety of each sample as input, but the gradient is computed stochastically on a fraction of the paths in the backpropagation pipeline.

A closely related class of problems, which can also benefit from backdrop, is optimization objectives defined via non-decomposable loss functions. The rank statistic, a differentiable approximation of the ROC AUC is an example of such loss functions (Herschtal & Raskutti (2004)). Here again, minibatch SGD optimization is not viable as the loss function cannot be well approximated over small batch sizes. Backdrop can also be applied in these problems to significantly improve generalization performance.

**Main contributions.** In this paper, (a) we establish a means for stochastic gradient optimization which does not require a modification to the forward pass needed to evaluate the loss function and is particularly useful for problems with non-trivial subsample structure or with non-decomposable loss. (b) We explore the technique empirically and demonstrate the significant gains that can be achieved using backdrop in a number of synthetic and real world examples. (c) We introduce the "multi-scale GP texture generator", a flexible synthetic texture generator with clear hierarchical subsample

structure which can be used as a benchmark to measure performance of networks and optimization tools in problems with hierarchical subsample structure. The source code for our implementation of backdrop and the multi-scale GP texture generator is available online.

## 1.1 RELATED WORK

**Non-decomposable loss optimization.** The study of optimization problems with complex non-decomposable loss functions has been the subject of active research in the last few years. A number of methods have been suggested including methods for online optimization (Kar et al. (2014); Kar et al. (2015); Narasimhan et al. (2015)), methods to solve problems with constraints (Narasimhan (2018)), and indirect plug-in methods (Narasimhan et al. (2014); Koyejo et al. (2014)). There are also methods that indirectly optimize specific performance measures (Dembczynski et al. (2011); Ye et al. (2012)). Our contribution is fundamentally different from the previous work on the subject. Firstly, our approach can be applied to any differentiable loss function and does not change based on the exact form of the loss. Secondly, implementation of backdrop is simple, does not require changing the network structure or the optimization technique and can therefore be used in conjunction with any other batch optimization method. Finally, backdrop optimization can be applied to a more general class of problems where the loss is decomposable but the samples have complex hierarchical subsample structure.

**Study of minibatch sizes.** Recently there have been many studies analyzing the effects of different batch sizes on generalization (Wilson & Martinez (2003); LeCun et al. (1998); Keskar et al. (2016); Masters & Luschi (2018)) and on computation efficiency and data parallelization (Das et al. (2016); Dean et al. (2012); Hoffer et al. (2017)). These studies, however, are primarily concerned with problems such as classification accuracy with cross-entropy or similar decomposable losses. There are no in-depth studies specifically targetting the effects of varying the batch size for non-decomposable loss functions that we are aware of. We address this in passing as part of one of our examples in Sec. 3.

## 2 BACKDROP STOCHASTIC GRADIENT DESCENT

Let us consider the following problem. Given dataset $S = \{x_1, \cdots, x_N\}$ with iid samples $x_n$, we wish to optimize an empirical loss function $L = L_\theta(S)$, where $\theta$ are parameters in a given model. We can employ gradient descent (GD), an iterative optimization strategy for finding the minimum of $L$ wherein we take steps proportional to $-\nabla_\theta L$. An alternate strategy which can result in better generalization performance is to use stochastic gradient descent (SGD), where instead we take steps according to $-\nabla_\theta L_n = -\nabla_\theta L(x_n)$, i.e. the gradient of the loss computed on a single sample. We can further generalize SGD by evaluating the iteration step on a number of randomly chosen samples called minibatches (Ian Goodfellow & Courville (2016); Bottou et al. (2018)). We cast GD and minibatch SGD as graphical models in Figs. 1a and 1b, where the solid and dashed lines respectively denote the forward and backward passes of the network and $N$, $B$, and $N/B$ denote the number of samples, the number of minibatches and the minibatch size.

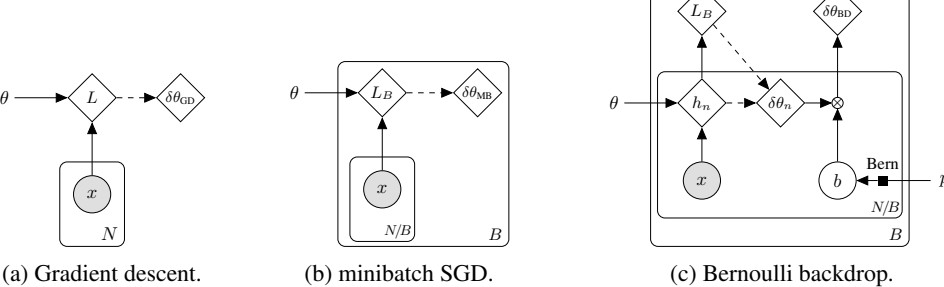

(a) Gradient descent.  (b) minibatch SGD.  (c) Bernoulli backdrop.

Figure 1: Graphical model representation of GD, minibatch SGD and backdrop with Bernoulli masking. The solid and dashed lines represent the forward pass and gradient computation respectively.

Now we consider two variations of this problem. First let us assume that the total loss we wish to optimize is not decomposable over the individual samples, i.e. it cannot be written as a sum of sample losses, for example it might depend on some global property of the dataset and include cross terms between different samples, i.e. $L = f(\vec{h})$, where $h_n$ is the hidden state computed from sample $x_n$ and $f$ is some non-decomposable function. There are many problems where these kinds of losses arise, and we discuss a few examples in Sec. 3. In extreme cases, the loss requires all N data points and batching the data is no longer meaningful (i.e. $B = 1$). In other cases, a large number of samples is required to accurately approximate the loss (i.e. $N/B \gg 1$) and hence minibatch SGD is also not optimal.

In order to deal with this problem and recover stochasticity to the gradient, we propose the following. During optimization, we evaluate the total loss during the forward pass, but for gradient calculation during the backward pass, we only compute $\delta\theta$ with respect to one (or some) of the samples. Note that if the loss $L$ can be decomposed to a sum of sample losses $L = \sum L_n$, this procedure would be identically equal to SGD. In practice, we implement backdrop by using a Bernoulli random variable to choose which samples to perform the gradient update with respect to (Fig. 1c). We call this method backdrop as it is reminiscent of dropout applied to the backpropagation pipeline.

We now consider a different variation of problem. Let us assume that each sample $x$ consists of a number of subsamples $x = \{\sigma_1, \cdots, \sigma_M\}$ where the subsamples $\sigma$ are no longer required to be iid. For example, an image can be considered as a collection of smaller patches, or a sequence can be considered as a collection of individual sequence elements. $\sigma_i$ can also be defined implicitly by the network, for example it could refer to the (possibly overlapping) receptor fields of the individual outputs of a convolutional layer. Similar to above, we take the sample loss to be a non-linear function of the subsamples i.e. $L_n = f(\vec{h})$ which again cannot be decomposed into a sum over the individual subsamples. We can implement backdrop by defining $\delta\theta_{\text{BD}}$ in a manner analogous to the previous case. Figure 2 depicts this implementation of backdrop on subsamples, where the $B$ and $N/B$ are as before and the $M$ plate represents the subsamples.

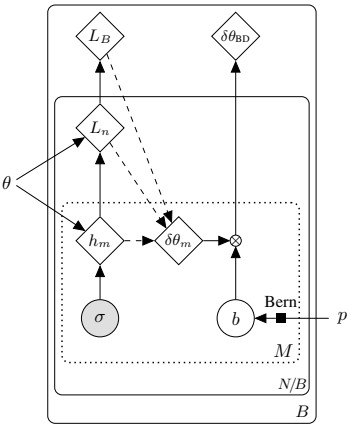

Figure 2: Backdrop on subsamples with Bernoulli masking. The dotted plate denotes the fact that the subsamples are generically not iid.

Note that while it is possible to think of the first class of problems with non-decomposable losses as a special case of the second class of problems with subsamples, we find that it is conceptually easier to consider these two classes separately.

## 2.1 Implementation

Since backdrop does not modify the forward pass of the network, it can be easily applied in most architectures without any modification of the structure. The simplest method to implement backdrop is via the insertion of a masking layer $M_p$ defined as follows. For vector $\vec{v} = (v_1, \cdots, v_N)$ we have:

$$M_p(v_n) \equiv v_n, \quad \nabla_{v_n} M_p(v_m) \equiv \delta_{nm} \frac{N}{|\vec{b}|_1} b_n (1 - p), \tag{1}$$

where $\vec{b}$ is a length $N$ vector of Bernoulli random variables with probability $1 - p$ and $|\vec{b}|_1$ counts the number of its nonzero elements. In words, $M_p$ acts as identity during the forward pass but drops a random number of gradients given by probability $p$ during gradient computation. The factor in front of the gradient is normalization such that the $\ell_1$ norm of the gradient is preserved.

For the problem where the total loss is a non-decomposable function of the sample latent variables $L = f(\vec{h})$, we can implement backdrop simply by inserting a masking layer between $h_n$ and $L$ (Fig. 1c):

$$L_{\text{BD}} = f(M_p(\vec{h})) \Rightarrow \delta\theta_{\text{BD}} = \nabla_\theta f(M_p(\vec{h})) = \frac{N}{|\vec{b}|_1} \sum_n b_n \nabla_\theta h_n \cdot \nabla_{h_n} f(\vec{h}), \tag{2}$$

resulting in only a fraction of the gradients being accumulated for the gradient descent update as desired. We would say backdrop in this case is masking its input along the minibatch direction $n$, resulting in a smaller *effective batch size* or EBS, which we define as EBS $\equiv N/B \times (1 - p)$.

In more complicated cases, for example if we want to implement backdrop on the subsamples as in Fig. 2, the location of the masking layer along the pipeline of the network may depend on the details of the problem. We will discuss this point with a number of examples in Sec. 3. However, even when the scale and structure of the subsamples is not known, it is possible to insert masking layers at multiple points along the network and treat the various masking probabilities as hyperparameters. In other words, it is not necessary to know the details of structure of the data in order to implement and take advantage of the generalization benefits provided by backdrop.

We note that while this implementation of backdrop is reminiscent to dropout (Hinton et al. (2012)), there are major differences. Most notably, the philosophy behind the two schemes is fundamentally different. Dropout is a network averaging method that simultaneously trains a multitude of networks and takes an averages of the outcomes to reduce over-fitting. Backdrop, on the other hand, is a tool designed to take advantage of the structure of the data and introduce stochasticity in cases where SGD is not viable. On a technical level, with backdrop, the masking takes place during the backward pass while the forward pass remains undisturbed. Because of this, backdrop can be applied in scenarios where the entire input is required for the task and it can also be simultaneously used with other optimization techniques. For example, it does not suffer the same incompatibility issues that dropout has with batch-normalization (Li et al. (2018)).

## 3 EXAMPLES

In this section, we discuss a number of examples in detail in order to clarify the effects of backdrop in different scenarios. We focus on the two problem classes discussed in Sec. 2, i.e. problems with non-decomposable losses and problems where we want to take advantage of the hierarchical subsample structure of the data. For our experiments, we use CIFAR10 (Krizhevsky et al. (2009)), the Ponce texture dataset (Lazebnik et al. (2005)) and a new synthetic dataset comprised of Gaussian processes with hierarchical correlation lengths. Note that in these experiments, the purpose is not to achieve state of the art performance, but to exemplify how backdrop can be used and what measure of performance gains one can expect. In each experiment we scan over a range of learning rates and weight decays. We train a number of models in each case and report the average of results for the configuration that achieves the best performance. The structure of the models used in the experiments is given in Tab. 1.

| Ponce | GP | CIFAR |
|---|---|---|
| $400\times300$ monochrome | $1024\times1024$ monochrome | $32\times32$ RGB image RGB |
| $7\times7$ conv 64 $2\times \begin{cases} 3\times3 \text{ conv 64 stride 2} \\ 3\times3 \text{ mp stride 2} \end{cases}$ | $4\times \begin{cases} 3\times3 \text{ conv 64} \\ 3\times3 \text{ conv 64} \\ 3\times3 \text{ mp stride 2} \end{cases}$ | $3\times3$ conv 96 $3\times3$ conv 96 $3\times3$ conv 96 stride 2 |
| | masking layer | |
| | $3\times \begin{cases} 3\times3 \text{ conv 64} \\ 3\times3 \text{ conv 64} \\ 3\times3 \text{ mp stride 2} \end{cases}$ $3\times3$ conv 64 $3\times3$ conv 4 | $2\times \{3\times3 \text{ conv 192}$ $3\times3$ conv 192 stride 2 $2\times \{3\times3 \text{ conv 192}$ $3\times3$ conv 10 |
| masking layer * | | |
| average-pool over remaining spatial dimensions | | |
| softmax | | |

Table 1: Network structures. With the exception of the final layer, all conv layers are followed by ReLU and batch-norm. The network used in CIFAR10 problems with non-decomposable loss do not employ the (*) masking layer. In these problems backdrop is implemented as in Eq. (4).

### 3.1 BACKDROP ON A NON-DECOMPOSABLE LOSS

There are an increasing number of learning tasks which require the optimization of loss functions that cannot be written as a sum of per-sample losses. Examples of such tasks include manifold learning, problems with constraints, classification problems with class imbalance, problems where the top/bottom results are more valued than the overall precision, or in general any problem which utilizes performance measures that require the the entire dataset for evaluation such as F-measures and area under the ROC curve and others.

As discussed in Sec. 2, SGD and minibatching schemes are either not applicable to these optimization problems or not optimal. We propose to approach this class of problems by using one or more backdrop masking layers to improve generalization (Fig 1c).

**Optimizing ROC AUC on CIFAR10.** As the first example in this class of problems we consider optimizing the ROC AUC on images from the CIFAR10 dataset. Specifically, we take the images of cats and dogs from this dataset. To make the problem more challenging and relevant for ROC AUC, we construct a 5:1 imbalanced dataset for training and test sets (e.g. 5000 cat images and 1000 dog images on the training set). In a binary classification problem, the ROC AUC measures the probability that a member of class 1 receives a higher score than a member of class 0. Since the ROC AUC is itself a non-differentiable function, we use the rank statistic, which replaces the non-differentiable Heaviside function in the definition of the ROC AUC with a sigmoid approximation, as our loss function (Herschtal & Raskutti (2004)).

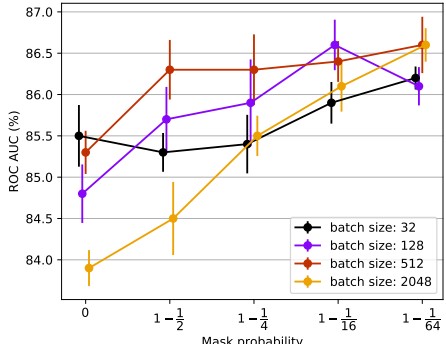

The results of this experiment are given in Fig. 3. We see that for all batch sizes, implementing backdrop improves the performance of the network. Moreover, the improvement that is achieved by implementing backdrop at batch size 2048 is greater than the improvement that we see when reducing batch size from 2048 to 32 without backdrop. This implies that in this problem, backdrop is a superior method of stochastic optimization than minibatching alone. For this experiment, performance deteriorates when reducing the batch size below 32, as the batch size becomes too small to approximate the rank statistic well. However, the effective batch size defined in Sec 2.1 has no such restriction.

Figure 3: ROC AUC vs. mask probability $p$ for different batch sizes.

**CIFAR10 with imposed latent space structure.** Consider again the image classification problem on CIFAR10 with a multi-class cross-entropy loss function. For demonstration purposes, we will try to impose a specific latent space structure by adding an extra non-decomposable term to the loss, demanding that the pair-wise $\ell^2$ distance of the per-class average of the hidden state be separated from each other by a fixed amount. This loss is similar in nature to disentangled representation learning problems (Lample et al. (2017); Whitney (2016); Ganin & Lempitsky (2014)).

If we denote the output of the final and next to final layers of the network on the $n$'th sample as $f_n$ and $v_n$, we can write the total loss as:

$$L = L_{XE}(\vec{f}) + L_{dist}(\vec{v}), \quad L_{dist}(\vec{v}) = + \sum_{c<c'} \left[ |\bar{v}_c - \bar{v}_{c'}|^2 - d^2 \right]^2, \quad \bar{v}_c = \mathbb{E}\left[ v \,|\, c \right] \quad (3)$$

where $c$ and $c'$ are different image classes and $\bar{v}_c$ is the class average of the hidden state. The total loss $L$ is no longer decomposable as a sum of individual sample losses. Furthermore, to make the problem more challenging, we truncate the CIFAR10 training set to a subset of 5500 images with an imbalanced number of samples per class. Specifically, on both train and test datasets, we truncate the $i$'th class down to $100i$ samples, i.e. 100 samples for class 1, 200 for class 2 and so on. We impose the same imbalance ratio on the test set.

Before proceeding to the results, it is important to note that we expect the two terms in the loss $L_{XE}$ and $L_{dist}$ to behave differently when optimized using minibatch SGD. The cross-entropy loss would

benefit from small batch sizes and its generalization performance is expected to suffer as the batch size is increased. However, as we will see, the second term becomes a poor estimator of the real loss when evaluated on minibatches of size smaller than 200. We would therefore expect a trade-off in the generalization performance of this problem if optimized using traditional methods.

To address this trade-off we use two masking layers with different masking probabilities for the two terms in the loss, i.e. we take the loss function:

$$L_{BD} = L_{XE}(M_{p_X}(\vec{f})) + L_{dist}(M_{p_D}(\vec{v})), \tag{4}$$

where $M_p$ are the backdrop masking layers defined in Eqs. (1) and (2). In this way we can benefit from two different effective batch sizes in a consistent manner and avoid the aforementioned trade-off between $L_{XE}$ and $L_{dist}$.

We train 10 models in each configuration given by batch sizes ranging from 32 to 2048 with $p_X$, $p_D$ ranging from 0 to 0.97. The results of this experiment are reported in Fig 4. Let us first consider the performance of the network without backdrop as a function of the minibatch size (Fig 4a purple lines). The solid lines denote $L_{dist}$ evaluated on the entire test dataset and the dashed lines are the average of $L_{dist}$ evaluated on minibatches, which we denote as $L_{dist}^{\text{MB}}$. Indeed we see that $L_{dist}^{\text{MB}}$ becomes smaller as we train the network with smaller minibatches, however, $L_{dist}$ remains roughly the same, implying that $L_{dist}^{\text{MB}}$ becomes a poorer estimator of $L_{dist}$ as batch sizes get smaller. As claimed, minibatch SGD does not benefit this optimization task.

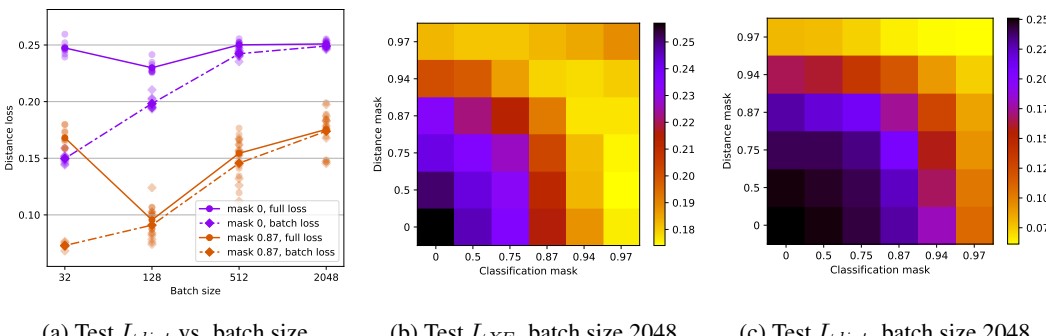

(a) Test $L_{dist}$ vs. batch size.  (b) Test $L_{XE}$, batch size 2048.  (c) Test $L_{dist}$, batch size 2048

Figure 4: The results of the CIFAR10 experiment with non-decomposable loss. The solid lines in (a) denote $L_{dist}$ and the dashed lines represent $L_{dist}^{\text{MB}}$, the average of $L_{dist}$ evaluated on minibatches.

Training the network with backdrop however, results in significant gains. In Figs 4b and 4c, we see the test results for $L_{XE}$ and $L_{dist}$ as a function of $p_X$ and $p_D$ i.e. the mask probabilities applied to $L_{XE}$ and $L_{dist}$ respectively (Eq. (4)). Note that increasing $p_X$ and $p_D$ generally reduces both losses but are more effective in reducing the loss function which they are directly masking. We note that at batch size 2048, the best performance of the network is achieved with masking probabilities $(p_X; p_D) = (0.94; 0.97)$.

## 3.2   BACKDROP ON SUBSAMPLES

The second class of problems where backdrop can have a significant impact are those where each sample has a hierarchical structure with multiple subsamples at each level of the hierarchy. There are many cases where this situation arises naturally. Image classification, time series analysis, translation and natural language processing all have intricate hierarchical subsample structure and we would like to be able to take advantage of this during training. In this section we provide three examples which demonstrate how backdrop takes advantage of this hierarchical subsample structure.

**Gaussian process textures.**   For our first example, we created a flexible synthetic texture dataset generator using Gaussian processes (GPs). The classes of any generated dataset are created by taking the convolution of a small-scale GP (5a) and a large-scale GP (5b) resulting in a 2-scale hierarchical structure (5c). In simple terms, each sample is comprised of a number of large blobs, each of which contains many much smaller blobs. The task is to learn to correctly classify the length scale of the GP fluctuations at each scale (details below.)

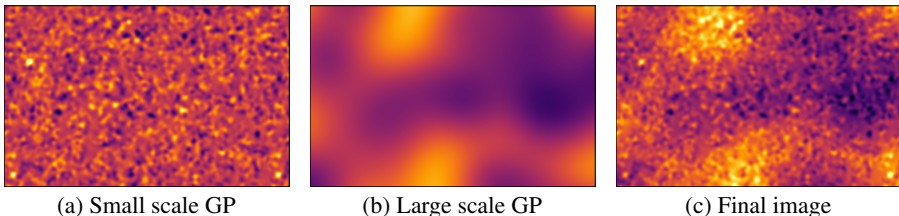

| (a) Small scale GP | (b) Large scale GP | (c) Final image |

Figure 5: Each sample in the two-scale GP dataset is the convolution of two GP processes with different scales.

Because of the hierarchical structure and the many realizations at each scale of the hierarchy, this dataset generator is the ideal test bed for our discussion. The generator also provides the flexibility to tune the difficulty of the problem via changing the individual correlation lengths as well as the number of subsample hierarchies.

A naive approach to problems with many realizations of the subsamples would be to crop each training sample into smaller images corresponding to the subsample sizes. In this multi-scale problem however, cropping is problematic. If we crop at the scale of the larger blobs, we are not fully taking advantage of the subsample structure of the smaller blobs. Whereas if we crop at the scale of the smaller blobs we destroy the large scale structure of the data. Furthermore, in many problems the precise subsample structure of the data is not known, making this cropping approach even harder to implement.

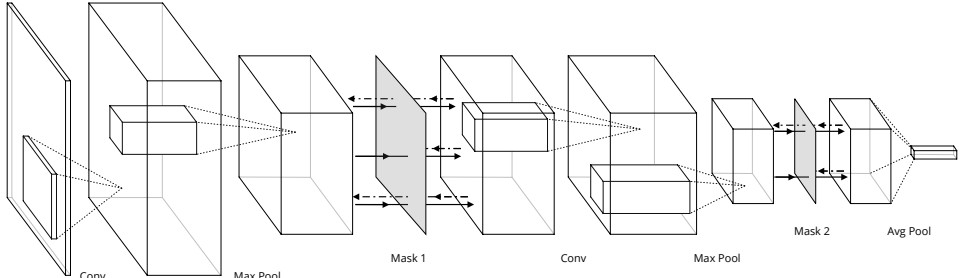

Figure 6: Cartoon of a masked convolutional network with masking layers at two different scales. The masking layers are transparent during the forward pass (solid lines) but block some percentage of the gradients during the backward pass (denoted by the dashed lines). The exact number of convolution, max pool and masking layers for each experiment is given in Tab. 1.

Backdrop provides an elegant solution to this problem. We utilize a convolutional network which takes the entirety of each image as input. In order to take advantage of the hierarchical subsample structure, we use two masking layers $M_{p_l}$ and $M_{p_s}$ respectively for the long and short range fluctuations. We insert these at positions along the network which correspond to convolution layers whose receptor field is roughly equal to the scale of the blobs (Fig. 6). This will result in a gradient update rule which updates the weights of the coarser feature detector convolutional layers according to a subset of the larger GP patches and will update the weights of the finer feature detector layers according to a subset of the small GP patches. $p_l$ and $p_s$ respectively determine how many of the large and small patches will be dropped during gradient computation. We therefore expect the classification accuracy of the different scales to change based on the two masking probabilities. Note that unlike cropping, this does not require knowing the exact size of the blobs. Firstly, since the network takes the entire image as an input, it will not lose classification power if we underestimate the size of the blobs. Furthermore, even if we have no idea about the structure of the data we can insert masking layers at more positions corresponding to a variety of receptor field sizes and use their respective masking probabilities as a hyper-parameter.

To highlight the improvements derived from backdrop masking, we treat this problem as a one-shot learning classification task, where each class has a single large training example. Each sample is a $1024 \times 1024$ monochrome image and the 4 different classes of the dataset have (small, large) GP scales corresponding to (9.5, 80), (10, 80), (9.5, 140) and (10, 140) pixels. The network is tasked with classifying each image into the correct class. The details of the network structure used are given in Tab. 1. In particular, the large masking layer acts on a $4 \times 4$ spatial lattice corresponding to patches

of size $256 \times 256$ and the small masking layer acts on a $64 \times 64$ lattice corresponding to patches of size $16 \times 16$. We train 10 models for masking probabilities $p_l \in \{0, 0.75, 0.94\}$, corresponding to keeping all, 4 or only one of the 16 large patches and $p_s \in \{0, 0.99, 0.999\}$ which correspond to keeping all, 40 or 4 of the $64^2$ small patches for gradient computation. We refrain from random cropping, flipping or any other type of data augmentation to keep the one-shot learning spirit of the problem where in many cases data augmentation is not possible.

The results of the experiment, evaluated on 100 test images can be seen in Tab. 2. We report the total classification accuracy, as well as the accuracy for correctly classifying each scale while ignoring the other. For reference, random guessing would yield 25% and 50% for total and scale-specific accuracies. We see that training with backdrop dramatically increases classification accuracy. Note that for networks trained with a single masking layer, classification is improved for the scale at which the masking is taking place, i.e. if we are masking the larger patches, the large scale classification is improved. The best performance is achieved with both masking layers present. However, we also see that having both masking layers at high masking probabilities can lead to a deterioration in classification performance. For comparison, we also provide the results of training the network with $512 \times 512$ and $256 \times 256$ fixed-location cropping augmentation, i.e. we respectively chop each training sample up into 4 or 16 equal sized images and train the network on these smaller sized images. With $512 \times 512$ cropping, the large scale discrimination power increases but this comes at the cost of the small scale discrimination power. The situation is worse with the tighter $256 \times 256$ crops, where the network is doing little better than random guessing.

| $(p_s, p_l)$ | Total | Small | Large |
|---|---|---|---|
| $(0, 0)$ | 45.9 | 64.7 | 74.1 |
| $(0, 0.75)$ | 54.1 | 67.4 | 82.9 |
| $(0, 0.94)$ | 49.1 | 55.3 | 88.2 |
| $(0.99, 0)$ | 57.1 | 77.9 | 73.8 |
| $(0.999, 0)$ | 57.4 | 68.2 | 84.1 |
| $(0.99, 0.75)$ | 53.5 | 55.0 | **98.2** |
| $(0.99, 0.94)$ | **66.2** | **78.7** | 84.6 |
| $(0.999, 0.75)$ | 54.7 | 61.5 | 88.8 |
| $(0.999, 0.94)$ | 55.0 | 58.5 | 95.0 |
| $512 \times 512$ crop | 45.3 | 50.9 | 87.9 |
| $256 \times 256$ crop | 34.3 | 53.9 | 55.3 |

Table 2: Classification accuracy of the small and large scales and overall accuracy and comparison with cropping.

**Ponce texture dataset.** The second example we consider is the Ponce texture dataset (Lazebnik et al. (2005)), which consists of 25 texture classes each with 40 samples of $400 \times 300$ monochrome images. In this case, the hierarchical subsamples are still present but the exact structure of the data is less apparent compared to the GPs considered above. For training, we use 475 samples keeping the remaining 525 samples as a test set. We use a convolutional neural net similar to the previous example but with a single backdrop layer which masks patches that are roughly $70 \times 60$ pixels in size. The details of the model are given in Tab. 1.

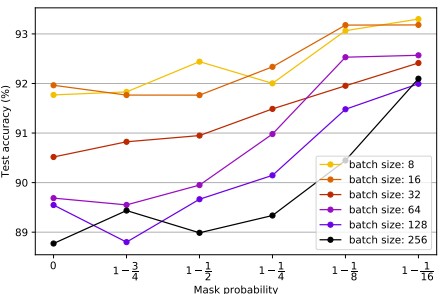

Figure 7: Test accuracy vs. mask probability $p$ for different batch sizes.

We train 25 models with batch sizes ranging from 8 to 256 and masking probabilities ranging from 0 to $93.75\%$ (roughly equivalent to keeping all the patches to keeping only 2 out of the 30 patches during gradient evaluation). The results of the experiment are reported in Fig. 7. For every value of the batch size, test performance improves as we increase the masking probability, but the gains are more pronounced for larger batch sizes. The best performance of the network is achieved by using small minibatches with high masking probability.

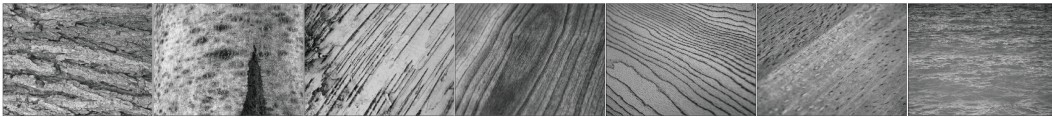

Figure 8: Samples from the Ponce texture dataset.

**CIFAR10.** Finally, we would like to demonstrate that even when the dataset does not have any apparent subsample structure, it is still possible to take advantage of generalization improvements provided by backdrop masking if we employ a network that implicitly defines such a structure. We demonstrate this using the task of image classification on the CIFAR10 dataset.

We employ the all-convolutional network structure introduced in Springenberg et al. (2014) with the slight modification that we use batch-normalization after all convolutional layers. The details of the model are given in Tab. 1. This model has a similar structure to the other models used in this section in that its output is a $6 \times 6$ classification heatmap (akin to semantic segmentation models) and the final classification decision is made by taking an average of this heatmap. The fact that this model works well (it achieves state of the art performance if trained with data augmentation), implies that each of the $6 \times 6$ points on the heatmap has a receptor field that is large enough for the purpose of classification. We can use this heatmap output of the network for implementing backdrop, in a similar manner to the previous two examples.

For this experiment, we use minibatch sizes from 16 to 2048 and backdrop mask probabilities from 0 to 0.9 (corresponding to keeping about 4 of the 36 points on the heatmap) and train 5 models for each individual configuration. We note that using backdrop in this situation leads to small and in some cases not statistically significant gains in test accuracy. However, as can be seen in Fig. 9, using backdrop can result in significant gains in test loss (up to $40\%$ in some cases), leading to the model making more confident classifications. The gains are especially pronounced for larger batch sizes. Note, however, that excessive masking for small batch sizes can deteriorate generalization.

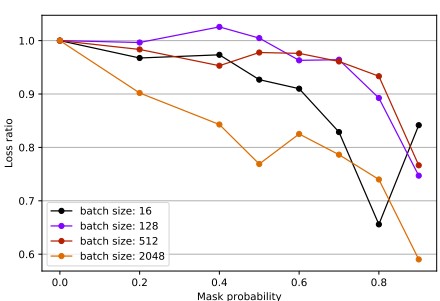

Figure 9: Ratio of test loss with backdrop to test loss without backdrop vs. mask probability $p$.

## 4  DISCUSSION

Backdrop is a flexible strategy for introducing data-dependent stochasticity into the gradient, and can be thought of as a generalization of minibatch SGD. Backdrop can be implemented without modifying the structure of the network, simply by adding masking layers which are transparent during the forward pass but drop randomly chosen elements of the gradient during the backward pass. We have shown in a number of examples how backdrop masking can dramatically improve generalization performance of a network in situations where minibatching is not viable. These fall into two categories, problems where evaluation of the loss on a small minibatch leads to a poor approximation of the real loss, e.g. problems with losses that are non-decomposable over the samples and problems with hierarchical subsample structure. We discussed examples for both cases and demonstrated how backdrop can lead to significant improvements. We also demonstrated that even in scenarios where the loss is decomposable and there is no obvious hierarchical subsample structure, using backdrop can lead to lower test loss and higher classification confidence. It would therefore be of interest to explore any possible effects of backdrop on vulnerability to adversarial attacks. We leave this venue of research to future work.

In our experiments, we repeatedly noticed that the best performance of the network, especially for larger batch sizes, is achieved when the masking probabilities of backdrop layers were extremely high (in some cases $> 98\%$). As a result, it takes more epochs of training to be exposed to the full information contained in the dataset, which can lead to longer training times. In our initial implementation of backdrop, the blocked gradients are still computed but are simply multiplied by zero. A more efficient implementation, where the gradients are not computed for the blocked paths, would therefore lead to a significant decrease in computation time as well as in memory requirements. In a similar fashion, we would expect backdrop to be a natural addition to gradient checkpointing schemes whose aim is to reduce memory requirements (Chen et al. (2016); Gruslys et al. (2016)).

Similar to other tools in a machine-learning toolset, backdrop masking should be used judiciously and after considerations of the structure of the problem. However, it can also be used in autoML schemes where a number of masking layers are inserted and the masking probabilities are then fine-tuned as hyperparameters.

The empirical success of backdrop in the above examples warrants further analytic study of the technique. In particular, it would be interesting to carryout a martingale analysis of backdrop as found in stochastic optimization literature (Gladyshev (1965); Robbins & Siegmund (1971)).

ACKNOWLEDGMENTS

We would like to thank Léon Bottou, Joan Bruna, Kyunghyun Cho and Yann LeCun for interesting discussions and input. We are also grateful to Kyunghyun Cho for suggesting the name backdrop. KC is supported through the NSF grants ACI-1450310 and PHY-1505463 and would like to acknowledge the Moore-Sloan data science environment at NYU. SG is supported by the James Arthur Postdoctoral Fellowship.

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
