# OpenReview forum: "Backdrop: Stochastic Backpropagation"
_ICLR.cc/2019/Conference_

### Official Review · AnonReviewer3 · 2018-11-02
**Interesting idea, lacking proper improvements and/or applications provided**

**Rating:** 5
**Confidence:** 3

**Review:**

This paper introduces a data dependent strategy to mask parts of the partial derivatives in the chain rule computation.

Typically with papers proposing modifications of the training regime of the neural network one would expect one of three outcomes:
 - a well justified, mathematically sound method, well tested in simple cases and with some proof of concept results on proper tasks
 - a more heuristic, empirical driven research, where strong results on proper tasks
 - method, however justified, allows us to do something previously impossible, removing some limitations/constraints (like biologically plausible learning etc.)

In its current form paper seems to lack any of these characteristics. On one hand method lacks any guarantees and on the other paper does not present significant improvements under any approved metrics, nor it introduces new which can be properly quantified. In fact, authors explicitly claim that empirical section "Note that in these experiments, the purpose is not to achieve state of the art performance, but to exemplify how backdrop can be used and what measure of performance gains one can expect.".

With methods like this it is almost obvious that resulting update is not an unbiased gradient estimator of any function. Consequently convergence/learning guarantees that we have for GD or SGD no longer apply. Do authors have any thoughts on how bad can it get? As noted in the text, other methods of "dropping" data (such as dropout) don't have this issue as they still estimate proper gradients. Here, since dropping is done inside the network only on backwards pass, resulting estimates could, in principle, lead to oscilations, divergence and other issues. If these are not encountered in practice it might be interesting to understand why.

If authors prefer to go through more empirical path, one would expect at least to see some baselines for tasks proposed, rather than comparing Backdrop to SGD. There are many methods that could be applied in scenarios like this, including dozens forms of dropout (which, as authors note, is not aimed at the same goals, but this does not mean that it will not shine under the metrics introduced, as they are non-standard and so - noone tested them in this exact regime).

I am happy to revisit my rating given authors restructure paper towards one of these paths (or other one which is not listed here).

---

### Official Review · AnonReviewer1 · 2018-11-02
**Small modification and not enough comparison to other methods**

**Rating:** 3
**Confidence:** 3

**Review:**

The authors propose to apply Dropout only in the backward pass, by applying a mask sampled from a Bernoulli distribution. They claim that this method can help in situations like optimizing non-decomposable losses where minibatch SGD is not viable.

First and foremost, the paper has an acknowledgement paragraph that gives information violating, in my sense, the anonymity requirement.

This being said, I have other concerns with the paper, and this possible violation didn't effect much my rating.

First, the authors claim that the proposed method "is a flexible strategy for introducing data-dependent stochasticity into the gradient". However, it doesn't seem to me that the sampled dropped nodes are data-dependent.

It is also not clear to me why the proposed method is better suited to non-decomposable losses and hierarchically structured data than the classical Dropout.

Moreover, while the method is clearly related to Dropout, the paper lacks of comparison to this regularizer.

This being said, the idea is sound, and can have a good impact in for example combining the good aspects of batch-normalization and dropout. However, the authors structured the paper on a completely different argument that doesn't convince me for the reasons cited above.

---

### Official Review · AnonReviewer2 · 2018-11-05
**The proposed Backdrop is similar to the traditional Dropout method. Overall this paper lacks of novelty and the observed generalization performance does not have convincing justification.**

**Rating:** 5
**Confidence:** 3

**Review:**

This paper proposes a stochastic based method, namely Backdrop, for updating the network structures via backpropagation type methods.  Backdrop inserts masking layers along the network; it acts as the identity in the forward pass, but as randomly masks parts of the backward gradient propagation. The paper claims this approach can significantly improves the overall generalization performance.

Although some difference to Dropout is summarized in Section 2, I still feel these two methods have almost the same idea, with just different implementation. Actually this Backdrop seems to have one more limitation in the parameter complexity, as it introduces several mask layers but keep the dense structures from other intermediate layers.

The proposed Backdrop uses Bernoulli distribution to select active variables. This is the very fundamental way in the conventional Dropout method. On the other hand, the authors do not  provide convincing justification how this can guarantee the improvement in subsequent generalization.

---

### Meta-Review · Area_Chair1 · 2018-12-13
**Limited novelty compared to Dropout**

**Confidence:** 4
**Recommendation:** Reject

**Metareview:**

Dear authors,

All reviewers pointed out that the proximity with Dropout warranted special treatment and that the justification provided in the paper was not enough to understand why exactly the changes were important. In its current state, this work is not suitable for publication to ICLR.

Should you decide to resubmit this work to another venue, please take the reviewers' comments into account.